# Plasma Phospholipid *n*-3/*n*-6 Polyunsaturated Fatty Acids and Desaturase Activities in Relation to Moderate-to-Vigorous Physical Activity through Pregnancy: A Longitudinal Study within the NICHD Fetal Growth Studies

**DOI:** 10.3390/nu12113544

**Published:** 2020-11-19

**Authors:** Liwei Chen, Yeyi Zhu, Zhe Fei, Stefanie N. Hinkle, Tong Xia, Xinyue Liu, Mohammad L. Rahman, Mengying Li, Jing Wu, Natalie L. Weir, Michael Y. Tsai, Cuilin Zhang

**Affiliations:** 1Department of Epidemiology, Fielding School of Public Health, University of California Los Angeles, Los Angeles, CA 90095, USA; cliwei86@ucla.edu (L.C.); xiatong@g.ucla.edu (T.X.); xinyue2396@g.ucla.edu (X.L.); 2Division of Research, Kaiser Permanente Northern California, Oakland, CA 94612, USA; yeyi.zhu@kp.org; 3Department of Biostatistics, Fielding School of Public Health, University of California Los Angeles, Los Angeles, CA 90095, USA; feiz@g.ucla.edu; 4Epidemiology Branch, Division of Intramural Population Health Research, *Eunice Kennedy Shriver* National Institute of Child Health and Human Development, National Institutes of Health, Bethesda, MD 20892, USA; stefanie.hinkle@nih.gov (S.N.H.); mengying.li@nih.gov (M.L.); jing.wu2@nih.gov (J.W.); 5Department of Population Medicine, Harvard Pilgrim Health Care Institute and Harvard Medical School, Boston, MA 02215, USA; mlr782@mail.harvard.edu; 6Department of Laboratory Medicine & Pathology, University of Minnesota, Minneapolis, MN 55455, USA; weirx065@umn.edu (N.L.W.); tsaix001@umn.edu (M.Y.T.)

**Keywords:** polyunsaturated fatty acids (PUFAs), physical activity, pregnant women

## Abstract

Maternal plasma phospholipid polyunsaturated fatty acids (PUFAs) play critical roles in maternal health and fetal development. Beyond dietary factors, maternal moderate-to-vigorous physical activity (MVPA) has been linked to multiple health benefits for both the mother and offspring, but studies investigating the influence of maternal MVPA on maternal PUFA profile are scarce. The objective of present study was to examine the time-specific and prospective associations of MVPA with plasma PUFA profile among pregnant women. This study included 321 participants from the National Institute of Child Health and Human Development (NICHD) Fetal Growth Studies–Singletons cohort. Maternal plasma phospholipid PUFAs and MPVA were measured at four visits during pregnancy (10–14, 15–26, 23–31, and 33–39 gestational weeks (GW)). Associations of maternal MVPA with individual plasma PUFAs and desaturase activity were examined using generalized linear models. Maternal MVPA was associated inversely with plasma phospholipid linoleic acid, gamma-linolenic acid, and Δ6-desaturase in late pregnancy (23–31 or 33–39 GW), independent of maternal age, race, education, parity, pre-pregnancy body mass index, and dietary factors. Findings from this longitudinal study indicate that maternal habitual MVPA may play a role on PUFAs metabolism, particular by alerting plasma *n*-6 subclass and desaturase activity in late pregnancy. These associations are novel and merit confirmation in future studies.

## 1. Introduction

Polyunsaturated fatty acids (PUFAs) are essential constitutes of cell membranes, particularly in the nervous and vascular system [1]. They also server as precursors for proinflammatory compounds called eicosanoids [2]. Emerging evidence has indicated that maternal circulating PUFA levels play important roles in achieving optimal maternal health and fetal development [3,4,5,6]. Since the fetal synthesis of PUFAs is very low, maternal circulating PUFA concentrations are critical resources for fetus [7]. Recently, there has been increased recognition that subclasses of PUFAs (i.e., *n*-3 and *n*-6 PUFAs) as well as individual PUFAs may have their unique biomedical properties and potentially divergent health effects. Indeed, maternal levels of *n*-3 PUFAs such as docosahexaenoic acid (DHA) and eicosadienoic acid (EPA) and *n*-6 PUFAs such as dihomo-gamma-linolenic acid (DGLA), docosatetraenoic acid (DTA), and arachidonic acid (AA) have been associated with pregnancy complications such as gestational diabetes (GDM) [8,9,10] and preeclampsia [11], as well as offspring birth weight [5,12,13] and childhood adiposity [14,15,16]. For example, a positive association between maternal plasma level of *n*-6 DGLA and GDM, but an inverse association between *n*-6 DTA and GDM was observed [10]. For the fetal outcomes, maternal levels of *n*-3 DHA and EPA were positively associated with birth weight, whereas *n*-6 DGLA and AA were inversely associated with birth weight [12,13]. These recent discoveries highlighted the importance of examining potentially modifiable factors that may influence maternal circulating levels of individual PUFAs.

Research on specific determinants of circulating individual PUFAs during pregnancy is still in its infancy. Circulating levels of PUFAs are functions of exogeneous and endogenous (i.e., de novo lipogenesis) sources. Previous data have illuminated roles of dietary intakes as exogeneous sources for plasma phospholipid PUFAs [17,18], whereas the role of physical activity (PA), another major modifiable behavioral factor with multiple health benefits, remains elusive [19,20,21]. PA, an important component of energy expenditure, has long been recognized as one of the major determinants of lipid metabolism [22]. Fatty acids are important sources of fuel for skeletal muscle, particularly during PA with moderate intensity and longer duration [23]. For example, a 6-month exercise training intervention was associated with reductions in concentrations of circulating free fatty acids and fatty acid by-products in middle-aged adults [24]. In an intervention study of 84 healthy nulliparous women, exercise intervention of a maximum of five sessions of 40-min aerobic exercise from 20 gestational week (GW) led to a lower level of circulating total free fatty acids at 35 GW compared to no exercise control [25]. However, there was no information on the impact of exercise on the individual FAs in that study. To date, we are not aware of any studies that have investigated whether maternal PA is associated with individual plasma PUFA levels and whether the association changes across pregnancy.

In the present study, we aimed to examine the association of maternal PA with individual and subclasses of plasma phospholipid PUFAs and PUFA desaturases activity at specific times across pregnancy among a cohort of medically low-risk pregnant women with singleton deliveries in the United States. Our study with longitudinal measures of maternal PA and plasma individual PUFAs allowed us to examine both time-specific and prospective relationships with the consideration of dynamic physiologic alterations during pregnancy.

## 2. Methods

### 2.1. Study Design and Population

We used data from the *Eunice Kennedy Shriver* National Institute of Child Health and Human Development (NICHD) Fetal Growth Studies-Singletons, a prospective cohort of 2802 healthy and racially/ethnically diverse pregnant women who had singleton pregnancies from 12 clinics across the United States between 2009 and 2013 [26]. Eligible participants were aged 18–40 years, had pre-pregnancy body mass index (BMI) that ranged from 19.0 to 44.9 kg/m^2^, were enrolled at between GW 8–13, and were free of preexisting conditions (i.e., pre-pregnancy hypertension under medical supervision, pre-pregnancy diabetes, renal or autoimmune disease, psychiatric disorders, cancer, or HIV/AIDs). The current study leveraged biomarker data in 321 women from a GDM nested case-control study (107 women with GDM and 214 women without GDM) who had plasma phospholipid PUFA profile measured throughout pregnancy. The GDM cases and non-GDM controls were matched at a ratio of 1:2 on maternal age (±2 years), race/ethnicity, and GW (±2 weeks) at blood collection. Following enrollment at 8–13 GW (visit 0), women were followed at 5 subsequent study visits throughout pregnancy: 16–22 GW (visit 1), 24–29 GW (visit 2), 30–33 GW (visit 3), 34–37 of weeks gestation (visit 4), and 38–41 weeks of gestation (visit 5). The Institutional Review Board approval (IRB number: 09-CH-N152) was obtained in May 2009 for all participating clinical sites, the data coordinating center, and NICHD.

### 2.2. Assessment and Quantification of Maternal Physical Activity

We applied the previously validated Pregnancy Physical Activity Questionnaire (PPAQ) [27] administered at 5 study visits: at 8–13 GW (visit 0) for measuring habitual maternal PA in the past 12 months (preconception and 1st trimester), and additional 4 times at subsequent study visits (16–22, 24–29, 30–33, and 34–37 GW) for measuring PA since the last visit. We asked women to report the amount of time they spent on different types of PAs. We calculated the time spent (minutes per week) on moderate to vigorous PA (MVPA) on the basis of the questions related to recreational PA or exercise, including walking, jogging, prenatal exercise classes, swimming, dancing, and others.

### 2.3. Biospecimen Preparation and Maternal PUFA Assessment

We collected venous blood samples at 4 study visits (i.e., visits 0, 1, 2, and 4, with no blood samples collection at visit 3) [26] on the basis of standard protocol. The actual times of blood sample collection were slightly different from the targeted times of study visits with a range of GW 10–14 at visit 0, 15–26 at visit 1, 23–31 at visit 2, and 33–39 at visit 4. To reduce the burden of study participants, we only collected the fasting sample at visit 1. The average time since the last meal was 3.68 h (SD: 3.86) at visit 0, 11.75 h (SD: 3.54) at visit 1, 3.19 h (SD: 3.98) at visit 2, and 2.90 h (SD: 3.71) at visit 4. Blood samples were immediately processed into EDTA plasma after collection and stored centrally at the NICHD repository at −80 °C until being shipped for biomarker analysis. We measured the plasma phospholipid PUFAs by a Hewlett Packard 5890 gas chromatography system in a certified clinical laboratory at the University of Minnesota (Minneapolis, MN) using a method developed as described in detail previously [10,28]. We identified 11 individual PUFAs with relative percentage above 0.05%, including 4 *n*-3 PUFAs (18:3*n*-3 (alpha-linolenic acid: ALA), 20:5*n*-3 (eicosapentaenoic acid: EPA), 22:5*n*-3 (*n*-3 docosapentaenoic acid: DPA), and 22:6*n*-3 (docosahexaenoic acid: DHA)) and 7 *n*-6 PUFAs (18:2*n*-6 (linoleic acid: LA), 18:3*n*-6 (gamma-linolenic acid: GLA), 20:2*n*-6 (eicosadienoic acid: EDA), 20:3*n*-6 (dihomo-gamma-linolenic acid: DGLA), 20:4*n*-6 (arachidonic acid: AA), 22:4*n*-6 (docosatetraenoic acid: DTA), and 22:5*n*-6 (*n*-6 docosapentaenoic acid: DPA)). The inter-assay coefficients of variation (CVs) for individual plasma phospholipid PUFAs were <11% for all [10]. We also estimated the plasma Δ5-desaturase (AA/DGLA) and Δ6-desaturase (GLA/LA) activities using the product-to-precursor ratios.

### 2.4. Covariates

We collected maternal sociodemographic characteristics, lifestyle factors, and reproductive and medical history from detailed questionnaires at visit 0 (8–13 GW). We calculated the pre-pregnancy body mass index (BMI) on the basis of height measured at enrollment and self-reported pre-pregnancy weight. We categorized pre-pregnancy BMI status as normal weight (<25.0 kg/m^2^), overweight (25.0–29.9 kg/m^2^), or obese (≥30.0 kg/m^2^). We assessed habitual dietary intakes for the last 3 months using the Food Frequency Questionnaire (FFQ) at visit 0 and using automated self-administered 24-h dietary recall (ASA24) at subsequent visits. Both dietary assessment tools were developed and validated by the National Cancer Institute, National Institutes of Health [29,30,31]. We calculated the Alternative Healthy Eating Index (AHEI) without alcohol on the basis of a method developed and validated by a previous study [32]. AHEI was developed as measure of dietary pattern and quality on the basis of foods and nutrients predictive of major chronic diseases such as diabetes, cardiovascular diseases, and cancers. Higher AHEI scores were associated with lower risk of these chronic diseases.

### 2.5. Statistical Analysis

The aim of current study was to examine the association between maternal PA and circulating levels of phospholipid PUFAs using biomarker data measured as part of a nested GDM case-control study. Because women with GDM were overrepresented in the analytic sample, we applied sampling weights to all analyses to represent the full NICHD Fetal Growth Studies—Singletons population and account for the oversampling of women with GDM [33]. Weights were created using the inverse probability of each subject in the full cohort (i.e., sampling probability of each non-GDM subject was calculated from a logistic regression in the full cohort, excluding GDM cases) [33]. In the weighted sample, 4% of women had GDM as opposed to 33% in the non-weighted sample. All analyses were implemented using SAS Version 9.4 (SAS Institute, Cary, NC, USA). Multiple testing was adjusted using Bonferroni correction with α = 0.05 as the level of significance.

We examined the distributions of maternal sociodemographic, anthropometric, reproductive, and lifestyle factors at visit 0. We then described the distributions of maternal PA and plasma phospholipid PUFAs at visits 0, 1, 2 and 4. We presented the descriptive statistics as weighted mean (standard error (SE)) for continuous variables and frequency (weighted percent) for categorical variables, if not mentioned otherwise. We compared the difference between the high and low levels of the MVPA using the cutoff of 150 min per week (low MVPA: <150 min/week; high MVPA: ≥150 min/week) according to recommendation by American College of Obstetricians and Gynecologists (ACOG) [34]. We used the *t*-test for comparing continuous variables and the chi-squared test for categorical variables, with both SEs and *p*-values for differences based on robust variance estimates.

For the primary analyses, we treated MVPA both as a continuous variable (total minutes spent on MVPA) and as a dichotomized variable (low and high MVPA as defined above). We performed generalized linear models with robust SE to examine the associations of maternal MVPA with plasma phospholipid PUFAs at each visit during pregnancy, adjusting for potential confounders, including maternal age (continuous), race/ethnicity (non-Hispanic White, non-Hispanic Black, Hispanic, Asian/Pacific Islander), education (high-school degree or less, associate degree, bachelor’s degree, or higher), marital status (married/living with a partner or not), insurance (private/managed care or Medicaid/other), pre-pregnancy BMI (continuous), dietary intake of total and individual PUFAs, and AHEI. We first examined the association of MVPA with total *n*-3 PUFAs, total *n*-6 PUFAs, and individual PUFAs and fatty acids ratios (as measures of desaturases activity) at each visit. Given the overall consistent results treating MVPA as a continuous or categorical variable, we presented the former as primary results. We also conducted a sensitivity analysis only among women without diagnosis of GDM since women with GDM might change their PA pattern following the recommendation of their physicians. Then, we examined the prospective associations of maternal MVPA in preconception and 1st trimester (i.e., assessed at visit 0) with plasma phospholipid PUFAs in early 2nd trimester (visit 1), late 2nd trimester (visit 2), and 3rd trimester (visit 4) in order to explore whether there was a long-term relationship of PA in pre-conception and 1st trimester with subsequent PUFAs. Additionally, we examined the prospective associations of MVPA in early 2nd trimester (visit 1) with plasma phospholipids PUFAs in late 2nd trimester (visit 2) and 3rd trimester (visit 4), as the 2nd trimester is considered the most comfortable period for PA during pregnancy [35].

To further examine the impact of change in maternal MVPA across pregnancy on plasma phospholipids PUFAs, we identified 4 groups of women by their levels of MVPA throughout the pregnancy: (1) consistently low (low-low group: low MVPA at both visits 0 and 4), (2) consistently high (high-high group: high MVPA at both visits 0 and 4), (3) increasing from low to high (low-high group: low MVPA at visit 0 and high MVPA at visit 4), and (4) decreasing from high to low (high-low group: high MVPA at visit 0 and low MVPA at visit 4). We also performed generalized linear models controlling for potential confounders to compare the individual PUFAs levels at visit 4 among these 4 MVPA groups with the low-low group as the reference group.

## 3. Results

### 3.1. Baseline Characteristics

When comparing the MVPA level in preconception and 1st trimester, women in the high-MVPA group (≥150 min/week) were older, more likely to be non-Hispanic Whites, be married/living with partners, have a bachelor’s degree or higher, be born in the United States, be nulliparous, have private or managed care insurance, and have drank alcohol 3 months before pregnancy, but were less likely to have smoked 6 months before pregnancy. Women with high MVPA also had a higher AHEI, but there were no statistically significant differences in intake of total energy and other nutrients such as total fatty acids, protein, carbohydrate, fiber, and cholesterol (Table 1).

### 3.2. Maternal PA and Plasma PUFAs at Each Visit

The median maternal total MVPA (minutes/week) and the percentage of women in the high MVPA group decreased over time during pregnancy. The median (range) MVPA was 135 (0–1005) min per week reported at visit 0, 75 (0–720) min per week at visit 1, 75 (0–675) min per week at visit 2, and 60 (0–645) min per week at visit 4. The percentages of women who had a high level of MVPA (≥150 min/week) were 46.7% at visit 0, 29.8% at visit 1, 29.8% at visit 2, and 19.6% at visit 4. In addition, 6.2%, 10.3%, 7.8%, and 13.4% of the women engaged zero MVPA at these visits, respectively.

At visit 0, sum of *n*-3 PUFAs and *n*-6 PUFAs accounted for 5.3% and 37.3% of total plasma phospholipid fatty acids, respectively. Among *n*-3 PUFAs, DHA (22:6*n*-3) was the most abundant form and accounted for 4.1% of total maternal plasma phospholipid fatty acids, followed by DPA (22:5*n*-3) for 0.67%, EPA (20:5*n*-3) for 0.29%, and ALA (18:3*n*-3) for 0.21%. LA (18:2*n*-6) was the most abundant form of *n*-6 PUFAs and accounted for 20.5% of total maternal plasma phospholipid fatty acids, followed by AA (20:4*n*-6) for 11.2%, DGLA (20:3*n*-6) for 3.5%, whereas other individual *n*-6 PUFAs contributed less than 1%. Individual plasma phospholipid PUFA concentrations and desaturase activities (estimated by PUFA ratios) also changed through pregnancy (Appendix A), with increases in the levels of ALA and LA and decreases in the levels of all others (i.e., EPA, DPA, GLA, DGLA, AA, Δ6-desaturase (GLA/LA), and Δ5-desaturase (AA/DGLA)).

### 3.3. Cross-Sectional and Prospective Associations of Maternal MVPA with Plasma PUFAs

Cross-sectionally, maternal MVPA (hours/week) was inversely associated with sum of plasma phospholipid *n*-6 PUFAs (β (SE) = −0.28 (0.07), % of total fatty acids; *p* < 0.001) and LA (β (SE) = −0.29 (0.08), % of total fatty acids; *p* < 0.001) at visit 2 and GLA (β = −0.004 (0.001), % of total fatty acids; *p* < 0.001) at visit 4, controlling for maternal age, race/ethnicity, education, marital status, parity, pre-pregnancy BMI, and AHEI. Further adjustment of insurance type, being born in the United States, or individual dietary factors (including individual dietary PUFA intake) did not change the results substantially. For example, with additional adjustment of dietary intake of LA, the association between maternal MVPA and plasma phospholipid LA became even stronger at visit 2 (β (SE) = −0.37 (0.08), % of total fatty acids; *p* < 0.001). No significant association was found of maternal MVPA with sum of *n*-3, sum of *n*-6, individual PUFAs, Δ5-desaturase activity, or Δ6-desaturase activity at visit 0 and visit 1. Compared to those in the low MVPA group, women in the high MVPA group had significantly lower levels of sum of *n*-6 PUFAs and LA at visit 2 (Table 2). 

In the sensitivity analysis among women without GDM, the overall patterns of the associations between maternal MVPA and plasma PUFAs were unchanged, whereas only the association with sum of *n*-6 PUFAs at visit 2 remained statistically significant after adjusting for multiple comparison (Appendix A).

The inverse association of maternal MVPA with LA was also observed from the prospective analysis—maternal MVPA at visit 0 was inversely associated with plasma phospholipid LA at both visit 2 (β (SE) = −0.19 (0.08); *p* = 0.02) and visit 4 (β (SE) = −0.35 (0.07); *p* < 0.001), and maternal MVPA at visit 1 was inversely associated with plasma phospholipid LA at visit 2 (β (SE) = −0.42 (0.11); *p* = 0.002) and visit 4 (β (SE) = −0.36 (0.10); *p* = 0.0005) (Table 3).

### 3.4. Change in Maternal MVPA across Pregnancy in Relation to Plasma PUFAs

Compared to women in the low-low MVPA group throughout the pregnancy (i.e., low MVPA at visit 0 and visit 4), women in the high-high MVPA group had a significantly higher adjusted level of *n*-3 DPA (β (SE) = 0.09 (0.03), % of total fatty acids; *p* = 0.002) at visit 4; women in the high-low MVPA group had significantly lower levels of *n*-3 EPA (β (SE) = −0.07 (0.02), % of total fatty acids; *p* = 0.003) and *n*-6 DTA (β (SE) = −0.13 (0.02), % of total fatty acids; *p* < 0.001); and women in the low-high MVPA group had significant lower levels of plasma *n*-6 GLA (β (SE) = −0.02 (0.01), % of total fatty acids; *p* = 0.001) and Δ6-desaturase activity (β (SE) = −0.001 (0.0004); *p* = 0.001), and higher Δ5-desaturase activity (β (SE) = 0.95 (0.33), *p* = 0.003) (Figure 1). In the sensitivity analysis in women without GDM, the overall patterns and the directions of the associations remained the same but were no longer statistically significant after adjusting for multiple testing (Appendix A).

## 4. Discussion

In this longitudinal study with measures of maternal MVPA and 11 individual plasma phospholipid PUFAs at four time points throughout pregnancy, we found both time-specific and prospective associations between MVPA and individual plasma phospholipid *n*-6 PUFAs and Δ6-desaturases. The directions of these associations varied by time and individual PUFAs. Our most important findings were the inverse associations of MVPA with plasma phospholipids LA, GLA, and Δ6-desaturases in late pregnancy (i.e., GW 24–29 and GW 34–37). These associations were consistent in both time-specific and prospective analyses and were independent of known influential factors for plasma PUFA concentrations. Additionally, pregnant women who increased their levels of MVPA from low to high between preconception and 1st trimester (GW 8–13) to late pregnancy (GW 34–37) had lower levels of GLA and Δ6-desaturase activity (estimated by GLA/LA), and a higher level of Δ5-desaturase activity (estimated by AA/LA).

Growing evidence has suggested that maternal circulating PUFA levels play important roles in achieving optimal maternal health and fetal development [3,4,5,6], with distinct associations varied by *n*-3 and *n*-6 PUFA subclasses and individual PUFAs. Identifying modifiable factors that can be targeted on interventions is critical for maternal and child health. Unfortunately, most previous studies have focused on dietary intake, mainly in Caucasian women, and did not take into consideration other factors. Research on the role of PA, another important modifiable lifestyle factor, on maternal circulation individual PUFAs is sparse. To our best knowledge, this study presents the first effort to comprehensively investigate the associations of maternal habitual MVPA with individual and subclasses of plasma phospholipid PUFA and desaturase activities at multiple time-points throughout pregnancy. We observed significant decreased Δ6-desaturase and increased Δ5-desaturase activities in late pregnancy among women who increased their MVPA from the preconception and 1st trimester to late pregnancy. This finding is also novel but consistent with results from non-pregnant population. The Δ5-desaturase and Δ6-desaturase are key enzymes that regulate the metabolism of PUFA in humans. Higher recreational PA from biking and sports were positively associated with Δ5-desaturase and inversely associated with Δ6-desaturase activity in the European Prospective Investigation into Cancer and Nutrition (EPIC) study, independent of BMI and dietary factors [36].

The inverse associations, from both time-specific and prospective analyses, between maternal MVPA and plasma *n*-6 PUFAs, particularly LA and GLA, are novel and interesting. We are not aware of any studies that have reported such association in either pregnant women or other populations. Therefore, it is difficult to make a direct comparison of our results with previous studies. Partly in line with our findings on total PUFA, in an intervention study of 84 healthy nulliparous women, exercise intervention with a maximum of five sessions of 40-min aerobic exercise from 20 GW led to a lower level of serum total free fatty acids at 35 GW compared to no exercise control [25]. However, there were no data about individual fatty acids in that study.

The biological mechanisms underlying the inverse relationship between maternal MVPA and plasma phospholipids *n*-6 PUFA such as LA and GLA are likely to be complicated. LA is an essential fatty acid to humans and its only resource is from dietary intake (primarily from nuts, seeds, and meats) [37]. Most of the LA obtained from diet is either stored in adipose tissues as energy or synthesized into other longer chain *n*-6 PUFAs in liver [38]. Although the plasma phospholipids LA level is responsive to recent dietary LA intake (a few weeks) [39], it is also influenced by its downstream PUFA biosynthesis pathway (i.e., elongation and β-oxidation to longer-chain PUFA) and uptakes by other tissues [40]. During this biosynthesis pathway, LA (18:2*n*-6) is first converted to GLA (18:3*n*-6) by Δ6-desaturase, which has been considered as the rate-limiting step. GLA, mainly an endogenous *n*-6 PUFA via de novo lipogenesis, is then rapidly elongated to DGLA (20:3*n*-6), which can be further desaturated to AA (20:4*n*-6) by Δ5-desaturase [41,42]. However, results from our study do not support the possibility that MVPA may influence the plasma phospholipids LA levels through enhancing its biosynthesis in liver since we also observed lower levels of GLA and Δ6-desaturase activity in women with higher levels of MVPA. A more reasonable explanation could be that MVPA increases tissue uptake of LA, likely by skeletal muscles and placenta. Fatty acids are an important source of fuel for skeletal muscle, particularly during PA with moderate intensity and longer duration [23]. Among pregnant women, exercise training may elicit adaptations in fatty acids partitioning to fetus, either directly through placental regulation of maternal metabolism or indirectly through regulation of maternal body composition changes [25]. The fetus cannot synthesize LA and ALA (i.e., the essential fatty acids), which can only be provided through placental transfer from the mother [43]. Indeed there is evidence suggesting that habitual exercise may be associated with higher activities of proteins/enzymes that enhance placental transfer of LA [44]. It is plausible that higher MVPA enhances the placental transfer of LA and/or other *n*-6 PUFA from maternal circulation into fetal circulation, subsequently resulting in lower LA in pregnant women.

We should acknowledge that inter-relationships among PUFA subclasses, individual *n*-6 PUFAs, and Δ5- and Δ6-desaturase activity are complicated and have not been fully elucidated. The observed associations of maternal MVPA with plasma individual *n*-6 PUFA and PUFA desaturase activities in this study should be interpreted with caution. It is possible that such associations may be influenced by complex inter-relationships among these individual PUFA and other unknown factors. However, we measured the relative concentrations of individual plasma phospholipid PUFAs as percent of total plasma fatty acids. The relative concentrations of individual PUFA reflect compositions of other individual PUFAs. As such, the reported relative concentration of PUFA reported in the present study has already accounted for relative concentrations of multiple individual PUFAs. The precise mechanisms that can explain the observed association are warranted in further investigations.

Our study has several notable strengths. First, this study is a prospective study with longitudinal data collection at four timepoints across pregnancy, thereby allowing the investigation of gestation-specific associations of maternal MVPA as well as changes in MVPA across pregnancy with individual plasma PUFA levels. Second, the study participants were diverse regarding the geographical and racial/ethnic distributions. Third, potential confounders during and prior to pregnancy were collected and available for controlling in the analyses. In particular, detailed dietary intakes were collected using validated instruments and at multiple times in pregnancy. Potential limitations of this study should be mentioned. Our self-reported physical activity and exercise may suffer from misclassification due to self-reporting or recall error. However, our method has been validated among pregnant women and widely applied in epidemiologic studies [27]. The percentage of women who meet the ACOG guideline for MVPA in our study was 30% in early–mid pregnancy (16–22 GW) and 20% in late pregnancy (34–37 GW). These number are highly consistent with what reported (31% at ~15 GW and 13% at 32–35 GW) in a study using accelerometer to measure PA objectively [45]. Moreover, women were only required to fast at visit 1. However, we measured PUFA levels in plasma phospholipids, which are commonly accepted to reflect the dietary intake of the past few days and are less likely to be influenced by dietary intakes of the last meal [46]. Furthermore, the time interval between the last meal and blood collection was not significantly associated with total PUFA, LA, GLA, or any individual PUFAs at visits with non-fasting plasma samples. The findings remained unchanged with additional adjusting for the time interval between the last meal and blood collection (Appendix A). Taken together, available evidence suggests that the non-fasting samples can be used in studies of lipid metabolism in pregnant women, but future studies with fasting blood samples in late pregnancy are needed to confirm our results.

## 5. Conclusions

In this longitudinal study among medically low-risk pregnant women with singleton deliveries in the United States, we found that MVPA was inversely associated with plasma phospholipid sum of *n*-6 PUFA, LA, GLA, and Δ6-desaturase, but positively associated with Δ5-desaturase in late pregnancy. These results are novel and, if confirmed, suggest that habitual MVPA may represent another modifiable factor beyond diet to improve plasma PUFA profile during pregnancy. Our study also supports the importance of the recent recognition of objectively measuring subclasses of plasma phospholipid PUFA (i.e., *n*-3 and *n*-6 PUFA) as well as individual plasma phospholipid PUFAs in research investigating their modifiable factors and health effects.

## Figures and Tables

**Figure 1 nutrients-12-03544-f001:**
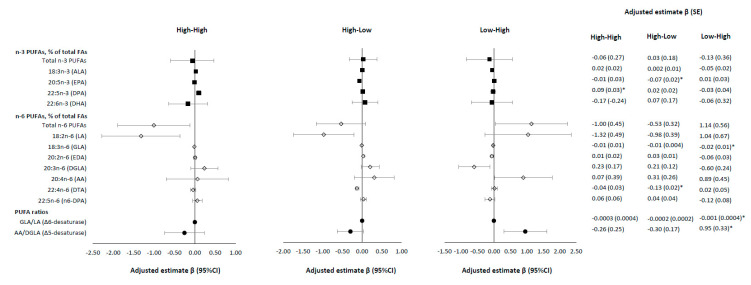
Changes in maternal moderate-to-vigorous physical activity (MVPA) levels between visit 0 (gestational weeks (GW): 8–13) and visit 4 (GW: 34–37) and plasma phospholipid polyunsaturated fatty acids (PUFAs) at visit 4 in the NICHD Fetal Growth Studies—Singleton Cohort. * *p*-value is significant after Bonferroni correction. Four groups by MVPA on the basis of their visit 0 and visit 4 data (High: MVPA ≥ 150 min/week; Low: MVPA < 150 min/week). High-High (High MVPA at visit 0 and High MVPA at visit 4, *N* = 40); High-Low (High MVPA at visit 0 and Low MVPA at visit 4, *N* = 90); Low-High (Low MVPA at visit 0 and High MVPA at visit 4; *N* = 15); Low-Low (Low MVPA at visit 0 and Low MVPA at visit 4, *N* = 136; reference group). Multivariable linear regression models adjusted for age (years), race, education, marriage status, nulliparous, and pre-pregnancy body mass index (BMI; kg/m^2^), and Alternative Health Eating Index (AHEI) score. Sampling weights were applied to all analyses to represent the full NICHD Fetal Growth Studies—Singletons population. Abbreviations: FAs, fatty acids; PUFAs, polyunsaturated fatty acids; AA, arachidonic acid; ALA, alpha-linolenic acid; DGLA, dihomo-gamma-linolenic acid; DHA, docosahexaenoic acid; DPA, docosapentaenoic acid; DTA, docosatetraenoic acid; EDA, eicosadienoic acid; EPA, eicosapentaenoic acid; GDM, gestational diabetes mellitus; GLA, gamma-linolenic acid; LA, linoleic acid.

**Table 1 nutrients-12-03544-t001:** Characteristics of pregnant women by moderate-to-vigorous physical activity (MVPA) at visit 0 (8–13 gestational weeks) in the National Institute of Child Health and Human Development (NICHD) Fetal Growth Studies—Singleton Cohort.

Characteristics	All (*N* = 321)	High MVPA (*N* = 150)	Low MVPA (*N* = 171)	*p*
Age, years	28.2 (0.3)	29.0 (0.5)	27.4 (0.4)	0.01
Race/ethnicity, *N* (%)				<0.001
Non-Hispanic Whites	75 (31.0)	52 (46.1)	23 (17.2)	
Non-Hispanic Blacks	45 (23.3)	13 (15.0)	32 (30.8)	
Hispanics	123 (27.2)	51 (22.4)	72 (31.5)	
Asian/Pacific Islanders	78 (18.5)	34 (16.5)	44 (20.4)	
Prepregnancy BMI, kg/m^2^	25.7 (0.3)	25.4 (0.4)	26.0 (0.4)	0.32
Prepregnancy BMI status, *N* (%)				0.28
Normal (BMI < 25.0, kg/m^2^)	156 (51.7)	78 (52.9)	78 (50.7)	
Overweight (BMI: 25.0–29.9 kg/m^2^)	99 (33.1)	44 (33.0)	55 (33.1)	
Obese (BMI > 30, kg/m^2^)	66 (15.2)	28 (14.1)	38 (16.2)	
Born in the United States, *N* (%)	182 (68.5)	94 (72.9)	88 (64.6)	<0.001
Education, *N* (%)				<0.001
High school or less	148 (45.5)	56 (39.8)	92 (50.7)	
Associates	50 (14.7)	21 (10.6)	29 (18.4)	
Bachelor’s or higher	123 (40.0)	73 (49.6)	50 (30.9)	
Insurance, *N* (%)				<0.001
Medicaid, other	108 (35.4)	37 (24.6)	71 (45.3)	
Private or managed care	211 (64.6)	112 (75.4)	99 (54.7)	
Married/living with a partner, *N* (%)	259 (72.9)	125 (84.1)	134 (62.7)	<0.001
Nulliparous, *N* (%)	144 (51.1)	78 (60.1)	66 (43.1)	<0.001
Smoked 6 months before pregnancy, *N* (%)	5 (0.7)	3 (0.23)	2 (1.1)	<0.001
Consumed alcoholic beverage 3 months before pregnancy, *N* (%)	198 (63.7)	106 (74.3)	92 (54.2)	<0.001
Dietary intakes ^§^,	(*N* = 191)	(*N* = 89)	(*N* = 102)	
Total energy, kcal/day	2176 (70.0)	2197 (90.2)	2156 (105.6)	0.77
Total carbohydrate, g/day	296 (10.6)	303 (14.2)	290 (15.6)	0.56
Total protein, g/day	85.6 (3.0)	87.6 (4.1)	83.8 (4.3)	0.52
Total fatty acids, g/day	77.4 (2.8)	76.4 (3.8)	78.2 (4.0)	0.75
Saturated fatty acids (SFAs), g/day	25.4 (1.0)	24.6 (1.4)	26.1 (1.5)	0.47
Monounsaturated fatty acids (MUFAs), g/day	29.5 (1.1)	29.7 (1.5)	29.4 (1.6)	0.89
Polyunsaturated fatty acids (PUFAs), g/day	16.3 (0.5)	16.0 (0.7)	16.6 (0.8)	0.60
Total dietary fiber, g/day	22.3 (0.8)	23.9 (1.0)	20.9 (1.3)	0.07
Cholesterol, mg/day	283 (11.8)	275 (15.5)	290 (17.5)	0.52
Alternative Heathy Eating Index (AHEI) score	44.1 (0.7)	46.5 (1.0)	41.9 (0.8)	<0.001

Data are presented as frequency and weighted percentage *N* (%) for categorical variables and weighted mean (standard errors, SE) for continuous variables. Sampling weights were applied to all analyses to represent the full NICHD Fetal Growth Studies—Singletons population. MVPA was assessed in the past 12 months. Women were grouped into high MVPA group if they had ≥ 150 min of moderate-to-vigorous physical activity (MVPA) per week. *p*-values were comparing between high MVPA vs. low MVPA group using *t*-tests for continuous variables and χ^2^-tests for categorical variables. ^§^ Dietary intakes were calculated among 198 women who completed the Food Frequency Questionnaires at visit 0. We additionally excluded seven women who had implausible total energy intake (i.e., <600 or >6000 kcal/day).

**Table 2 nutrients-12-03544-t002:** Time-specific associations of maternal moderate-to-vigorous physical activity (MVPA) and plasma phospholipid polyunsaturated fatty acids (PUFAs) at each study visits in the NICHD Fetal Growth Studies–Singleton Cohort.

	Visit 0 ^§^(8–13 Weeks)	Visit 1 ^  ^(16–22 Weeks)	Visit 2 ^  ^(24–29 Weeks)	Visit 4 ^  ^(34–37 Weeks)

	β (SE)	*p*	β (SE)	*p*	β (SE)	*p*	β (SE)	*p*
**MVPA as continuous variable (hour per week)**
**Sum of *n*-3 PUFAs** (% of total fatty acids)	0.004 (0.027)	0.90	−0.050 (0.038)	0.19	0.002 (0.038)	0.96	−0.002 (0.043)	0.96
18:3*n*-3 (ALA)	−0.00004 (0.001)	0.98	−0.0004 (0.002)	0.51	−0.006 (0.003)	0.03	0.0007 (0.003)	0.81
20:5*n*-3 (EPA)	0.005 (0.004)	0.21	0.002 (0.003)	0.41	0.002 (0.005)	0.73	−0.003 (0.005)	0.55
22:5*n*-3 (DPA)	0.001 (0.005)	0.80	−0.011 (0.006)	0.04	0.007 (0.006)	0.21	0.012 (0.005)	0.02
22:6*n*-3 (DHA)	−0.002 (0.025)	0.93	−0.042 (0.035)	0.22	−0.001 (0.034)	0.97	−0.012 (0.039)	0.76
**Sum of *n*-6 PUFAs** (% of total fatty acids)	−0.122 (0.058)	0.03	0.009 (0.061)	0.88	**−0.284 (0.068)**	**<0.001 ***	−0.091 (0.080)	0.26
18:2*n*-6 (LA)	−0.087 (0.061)	0.15	−0.071 (0.074)	0.34	**−0.287 (0.081)**	**<0.001 ***	−0.177 (0.094)	0.06
18:3*n*-6 (GLA)	−0.00002 (0.001)	0.98	0.002 (0.001)	0.04	0.001 (0.001)	0.48	**−0.004 (0.001)**	**<0.001 ***
20:2*n*-6 (EDA)	0.0001 (0.002)	0.94	−0.0002 (0.003)	0.95	−0.004 (0.002)	0.13	−0.001 (0.003)	0.83
20:3*n*-6 (DGLA)	0.003 (0.019)	0.88	0.013 (0.021)	0.53	−0.033 (0.021)	0.11	0.027 (0.028)	0.33
20:4*n*-6 (AA)	−0.032 (0.043)	0.45	0.051 (0.054)	0.35	0.024 (0.057)	0.68	0.050 (0.065)	0.44
22:4*n*-6 (DTA)	−0.005 (0.004)	0.28	0.002 (0.003)	0.58	−0.005 (0.006)	0.41	0.001 (0.006)	0.88
22:5*n*-6 (*n*6-DPA)	−0.002 (0.005)	0.75	0.012 (0.007)	0.07	0.019 (0.007)	0.01	0.012 (0.010)	0.22
**PUFA ratios**								
GLA/LA (Δ6-desaturase)	0.00002 (0.00004)	0.60	0.0001 (0.0004)	0.03	0.0001 (0.0005)	0.11	−0.0002 (0.00006)	0.01
AA/DGLA (Δ5-desaturase)	−0.009 (0.027)	0.73	−0.008 (0.029)	0.79	0.030 (0.028)	0.29	−0.025 (0.039)	0.52
**MVPA high vs. low (reference group)**
**Sum of *n*-3 PUFAs** (% of total fatty acids)	−0.28 (0.14)	0.05	−0.31 (0.17)	0.06	0.25 (0.18)	0.18	−0.09 (0.19)	0.66
18:3*n*-3 (ALA)	−0.004 (0.01)	0.64	0.01 (0.01)	0.41	−0.01 (0.01)	0.52	0.01 (0.01)	0.31
20:5*n*-3 (EPA)	−0.03 (0.02)	0.11	0.0001 (0.01)	0.99	0.06 (0.02)	0.02	0.02 (0.02)	0.42
22:5*n*-3 (DPA)	−0.03 (0.02)	0.26	−0.06 (0.03)	0.01	0.05 (0.03)	0.05	0.02 (0.02)	0.35
22:6*n*-3 (DHA)	−0.22 (0.13)	0.10	−0.26 (0.15)	0.10	0.15 (0.16)	0.80	−0.14 (0.18)	0.43
**Sum of *n*-6 PUFAs** (% of total fatty acids)	−0.31 (0.31)	0.32	0.09 (0.27)	0.73	**−1.37 (0.33)**	**<0.001 ***	−0.46 (0.37)	0.21
18:2*n*-6 (LA)	−0.24 (0.32)	0.45	−0.10 (0.33)	0.76	**−1.53 (0.39)**	**<0.001 ***	−0.44 (0.43)	0.31
18:3*n*-6 (GLA)	−0.001 (0.004)	0.69	0.01 (0.004)	0.06	0.01 (0.004)	0.05	−0.01 (0.01)	0.12
20:2*n*-6 (EDA)	−0.001 (0.01)	0.69	−0.001 (0.01)	0.97	−0.03 (0.01)	0.02	−0.01 (0.01)	0.30
20:3*n*-6 (DGLA)	0.06 (0.10)	0.52	0.13 (0.09)	0.16	−0.13 (0.10)	0.18	0.04 (0.13)	0.77
20:4*n*-6 (AA)	−0.13 (0.23)	0.55	−0.02 (0.24)	0.95	0.23 (0.28)	0.41	−0.09 (0.30)	0.77
22:4*n*-6 (DTA)	−0.04 (0.02)	0.08	0.001 (0.02)	0.94	0.02 (0.03)	0.60	0.01 (0.03)	0.66
22:5*n*-6 (n6-DPA)	0.05 (0.03)	0.05	0.07 (0.03)	0.02	0.07 (0.03)	0.04	0.04 (0.04)	0.42
**PUFA ratios**								
GLA/LA (Δ6-desaturase)	−0.00004 (0.0002)	0.86	0.0004 (0.0002)	0.06	0.001 (0.0002)	0.004	−0.0003 (0.0003)	0.32
AA/DGLA (Δ5-desaturase)	−0.15 (0.14)	0.28	−0.15 (0.13)	0.24	0.21 (0.14)	0.12	−0.05 (0.18)	0.78

Multivariable linear regression models adjusted for age (years), race, education, marital status, nulliparous, and pre-pregnancy body mass index (BMI; kg/m^2^) and Alternative Health Eating Index (AHEI) score ^±^. Individual PUFA is measured as % of total fatty acids. Sampling weights were applied to all analyses to represent the full NICHD Fetal Growth Studies–Singletons population. At visits 2 and 4, plasma phospholipid PUFAs were measured for all gestational diabetes (GDM) cases but only for one of the two controls. Thus, a different sample weight was applied for visits 2 and 4 to account this design. The bold numbers are statistically significant at alpha of 0.05 levels. * *p*-value is significant after Bonferroni correction. ^§^ Physical activity in the previous year. ^

^ Physical activity since the last visit. ^±^ Missing values in AHEI were imputed by mean of each visit. Abbreviations: PUFAs, polyunsaturated fatty acids; AA, arachidonic acid; ALA, alpha-linolenic acid; DGLA, dihomo-gamma-linolenic acid; DHA, docosahexaenoic acid; DPA, docosapentaenoic acid; DTA, docosatetraenoic acid; EDA, eicosadienoic acid; EPA, eicosapentaenoic acid; GDM, gestational diabetes mellitus; GLA, gamma-linolenic acid; LA, linoleic acid.

**Table 3 nutrients-12-03544-t003:** Prospective associations of maternal moderate-to-vigorous physical activity (MVPA) and plasma phospholipid polyunsaturated fatty acids (PUFAs) in the NICHD Fetal Growth Studies—Singleton Cohort.

	MVPA Visit 0 (8–13 Weeks)						MVPA Visit 1 (16–22 Weeks)			
	PUFAs Visit 1 (16–22 Weeks)		PUFAs Visit 2 (24–29 Weeks)		PUFAs Visit 4 (34–37 Weeks)		PUFAs Visit 2 (24–29 Weeks)		PUFAs Visit 4 (34–37 Weeks)	
	β (SE)	*p*	β (SE)	*p*	β (SE)	*p*	β (SE)	*p*	β (SE)	*p*
**Sum of *n*-3 PUFAs**	0.041 (0.031)	0.18	0.066 (0.038)	0.09	0.063 (0.035)	0.07	0.088 (0.053)	0.09	0.079 (0.049)	0.10
18:3*n*-3 (ALA)	0.002 (0.002)	0.32	−0.003 (0.003)	0.28	−0.002 (0.002)	0.39	−0.010 (0.004)	0.01	−0.003 (0.003)	0.38
20:5*n*-3 (EPA)	0.002 (0.002)	0.44	−0.001 (0.005)	0.91	−0.001 (0.004)	0.76	0.009 (0.007)	0.21	0.012 (0.006)	0.03
22:5*n*-3 (DPA)	0.003 (0.005)	0.45	0.007 (0.006)	0.21	0.004 (0.004)	0.33	0.015 (0.008)	0.06	0.010 (0.006)	0.11
22:6*n*-3 (DHA)	0.035 (0.028)	0.22	0.063 (0.034)	0.07	0.062 (0.032)	0.05	0.075 (0.047)	0.11	0.060 (0.044)	0.17
**Sum of *n*-6 PUFAs**	0.047 (0.050)	0.35	−0.034 (0.070)	0.62	−0.180 (0.063)	0.004	−0.221 (0.096)	0.02	−0.199 (0.090)	0.03
18:2*n*-6 (LA)	0.0004 (0.061)	0.99	−0.190 (0.084)	0.02	**−0.348 (0.073)**	**<0.001 ***	**−0.419 (0.113)**	**0.0002 ***	**−0.358 (0.103)**	**0.001 ***
18:3*n*-6 (GLA)	−0.00002 (0.001)	0.98	0.002 (0.001)	0.04	0.003 (0.001)	0.01	0.002 (0.001)	0.23	0.0001 (0.001)	0.93
20:2*n*-6 (EDA)	0.002 (0.003)	0.37	0.003 (0.002)	0.28	0.0002 (0.002)	0.92	-0.005 (0.003)	0.13	−0.008 (0.003)	0.01
20:3*n*-6 (DGLA)	−0.014 (0.017)	0.42	0.028 (0.021)	0.18	0.043 (0.022)	0.05	−0.004 (0.029)	0.90	−0.00004 (0.031)	1.00
20:4*n*-6 (AA)	0.058 (0.044)	0.19	0.120 (0.057)	0.03	0.116 (0.051)	0.02	0.185 (0.079)	0.02	0.147 (0.071)	0.04
22:4*n*-6 (DTA)	0.008 (0.003)	0.004	−0.003 (0.006)	0.62	−0.007 (0.005)	0.16	−0.005 (0.008)	0.51	−0.003 (0.007)	0.68
22:5*n*-6 (*n*6-DPA)	−0.008 (0.005)	0.14	0.005 (0.007)	0.48	0.014 (0.008)	0.09	0.025 (0.010)	0.01	0.023 (0.011)	0.03
**PUFA ratios**										
GLA/LA (Δ6-desaturase)	−0.000002 (0.00004)	0.96	0.0001 (0.0001)	0.02	**0.0002 (0.00005)**	**<0.001 ***	0.0002 (0.0001)	0.03	0.00009 (0.00007)	0.19
AA/DGLA (Δ5-desaturase)	0.032 (0.023)	0.17	0.010 (0.028)	0.71	−0.036 (0.031)	0.25	0.038 (0.039)	0.33	0.003 (0.044)	0.94

Multivariable linear regression models adjusted for age (years), race, education, marital status, nulliparous, and pre-pregnancy body mass index (BMI; kg/m2), and Alternative Health Eating Index (AHEI) score ^±^. Individual PUFA is measured as % of total fatty acids. Sampling weights were applied to all analyses to represent the full NICHD Fetal Growth Studies–Singletons population. At visits 2 and 4, plasma phospholipid PUFAs were measured for all GDM cases but only for one of the two controls. Thus, a different sample weight was applied for visits 2 and 4 to account for this design. The bold numbers are statistically significant at alpha of 0.05 levels. * *p*-value is significant after Bonferroni correction. ^±^ Missing values in AHEI were imputed by mean of each visit. Abbreviations: PUFA, polyunsaturated fatty acids; AA, arachidonic acid; ALA, alpha-linolenic acid; DGLA, dihomo-gamma-linolenic acid; DHA, docosahexaenoic acid; DPA, docosapentaenoic acid; DTA, docosatetraenoic acid; EDA, eicosadienoic acid; EPA, eicosapentaenoic acid; GDM, gestational diabetes mellitus; GLA, gamma-linolenic acid; LA, linoleic acid.

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
