# Peer review of "Plasma Phospholipid n-3/n-6 Polyunsaturated Fatty Acids and Desaturase Activities in Relation to Moderate-to-Vigorous Physical Activity through Pregnancy: A Longitudinal Study within the NICHD Fetal Growth Studies"

_nutrients, 2020, doi:10.3390/nu12113544_

Round 1

Reviewer 1 Report

The manuscript submitted for the review is a valuable contribution to both the science and practice.

I have some minor remarks that should be taken into account:

PUFAs instead of PUFA throught the mnuscript

Introduction lines 47-50: "maternal levels" - what levels, too high or too low? "associated with pregnancy complications" - in what way associated, excess of deficiency of n-3 and/or n-6? I think the authors should briefly prsent these issues.

line 77: "were selected from"

lines 102-103: why bood samples wwere not collected at 3rd visit?

line: 109: explain ALA abbreviation

line 111: explain LA

line 113: explain DPA

Table 1: Multiparous instead of Nulliparous?

Saturated fatty acids (SFAs) not SAT

MUFAs instead of MUFA

line 192: additionally instead of additional

line 210 and further "i.e.," instead of "i.e."

Author Response

The manuscript submitted for the review is a valuable contribution to both the science and practice. I have some minor remarks that should be taken into account:

PUFAs instead of PUFA throught the manuscript

Re: Thank you and we have used the PUFAs in the revised manuscript.

Introduction lines 47-50: "maternal levels" - what levels, too high or too low? "associated with pregnancy complications" - in what way associated, excess of deficiency of n-3 and/or n-6? I think the authors should briefly prsent these issues.

Re: Thanks for this comment. Since the directions of the associations between PUFAs and pregnancy outcomes are varied by individual PUFAs, we have provided some examples in the revised manuscript (page 2, line 51-53).

“For example, a positive association between maternal plasma level of n-6 DGLA and GDM, but an inverse association between n-6 DTA and GDM was observed.  For the fetal outcomes, maternal levels of n-3 DHA and EPA were positively associated with birth weight, whereas n-6 DGLA and AA were inversely associated with birth weight”

line 77: "were selected from"

Re: Thank you but the study participants were from the NICHD Fetal Growth Studies-Singletons cohort. We didn’t “select” them. We have revised the manuscript to avoid the confusion. (page 2, line 79)

“We used data from the Eunice Kennedy Shriver National Institute of Child Health and Human Development (NICHD) Fetal Growth Studies”

lines 102-103: why bood samples wwere not collected at 3rd visit?

Re: The NICHD Fetal Growth Study was designed as a prospective study with longitudinal data collection more than 5 times over pregnancy. Compared with data collection on other information, blood collection and processing are always costly and time consuming and impose more extra burden to study participants.  As such, we didn’t collect blood samples at each visit, for example, visit 3, but made sure to have blood collected at least once in each trimester of pregnancy.  

line: 109: explain ALA abbreviation

Re: Thank you and we have added the full name of ALA in the revised manuscript.

line 111: explain LA

Re: Thank you and we have added the full name of LA in the revised manuscript.

line 113: explain DPA

Re: Thank you and we have added the full name of DPA in the revised manuscript.

Table 1: Multiparous instead of Nulliparous?

Re: We think using Nulliparous is appropriate. 

Saturated fatty acids (SFAs) not SAT

Re: Thank you and we have corrected this typo in the revised manuscript.

MUFAs instead of MUFA

Re: Thank you and we have used MUFAs in the revised manuscript.

line 192: additionally instead of additional

Re: Thank you and we have corrected this in the revised manuscript.

line 210 and further "i.e.," instead of "i.e."

Re: Thank you and we have used “i.e.,” in the revised manuscript.

Reviewer 2 Report

This paper shed new lights about the effect of physical activity on lipid profile in pregnancy setting. Nevertheless, there are some methodological aspects and littles errors to consider and to revise:

Line 54 Please, the correct word is "de novo lipogenesis", not “do novo”.

Line 82, The authors have inserted the acronym definition at line 64, please maintain only GW.

Line 83, Word 'rental' is a typo' maybe renal?

Line 105, please detail the processing method in section 2.3.

Line 126, Please detail AHEI briefly.

Line 359-361, This author's sentence is not supported by reference [47]. In the review article of Arab, the author had written ' Other serum or plasma measures reflect the dietary intakes of the past few hours (triglyceride) or the past few days (cholesterol ester and phospholipid fatty acids).'. In addition, he had written 'The next most immediate biomarker medium is the serum or plasma levels of individual fatty acids, which can reflect intake over the last few days or meals.' For your knowledge, check this paper: "Am J Clin Nutr 2007;86:74–81".

Lines 361-365, the observations about fasting and non-fasting condition in subjects analyzed is not completely supported by reference [48]. In the cited paper the authors compare lipid in the fasting state and after 1 or 2 hours after breakfast, but in your works the biospecimens preparation, in particular, the time of last meal of subjects is not provided in the sections 2.3. The types of meal (breakfast or lunch or dinner) are very different in the amount and composition of lipids. Also, 2 hours is too less to observe a change in lipid profile after a meal, because the lipid digestive process lasts 4-6 hours approximately.

In the discussion highlight that estimation of the plasma delta-5-desaturase (AA/DGLA) and delta-6-desaturase (GLA/LA) activities using the product-to-precursor ratios is a study limit, because not exclude the effects of dietary intake in the previous day before collecting sample plasma, because 24h dietary recall is not performed at collecting blood time.

The discussion would be corrected with the previous suggestions.

Technical additional suggestions:

  1. Please improve the sharpness and quality of Figure 1.
  2. Supplement Table 3 are not present in supplementary material attached.

Author Response

This paper shed new lights about the effect of physical activity on lipid profile in pregnancy setting. Nevertheless, there are some methodological aspects and littles errors to consider and to revise: 

Line 54 Please, the correct word is "de novo lipogenesis", not “do novo”.

Re: Thank you and we have corrected this typo in the revised manuscript.

Line 82, The authors have inserted the acronym definition at line 64, please maintain only GW.

Re: Thank you and we have used “GW” in line 82 in the revised manuscript. 

Line 83, Word 'rental' is a typo' maybe renal?

Re: Thank you and we have corrected this typo in the revised manuscript. 

Line 105, please detail the processing method in section 2.3. 

Re: Thank you and we have added more details (see below) regarding the blood samples processing method (page 3).

“To reduce the burden of study participants, women were only required to fast at visit 1 (an overnight fasting of 8-14 hours).  The average time since the last meal was 3.68 hours (SD: 3.86) at visit 0, 11.75 hours (SD: 3.54) at visit 1, 3.19 hours (SD: 3.98) at visit 2, and 2.90 hours (SD: 3.71) at visit 4. Blood samples were immediately processed into EDTA plasma”

“Blood samples were immediately processed into EDTA plasma after collection and stored centrally at the NICHD repository at -80â—‹ C until being thawed immediately before assay according to a standardized protocol. We measured the plasma phospholipid PUFAs by a Hewlett Packard 5890 gas chromatography system in a certified clinical laboratory at University of Minnesota (Minneapolis, MN) using a method developed as described in detail previously.”

Line 126, Please detail AHEI briefly.

Re: Thank you and we have provided more information regarding the AHEI in the revised manuscript (page 3). 

“AHEI was developed as measure of dietary pattern and quality based on foods and nutrients predictive of major chronic diseases such as diabetes, cardiovascular diseases, and cancers. Higher AHEI scores were associated with lower risk of these chronic diseases.”

Line 359-361, This author's sentence is not supported by reference [47]. In the review article of Arab, the author had written ' Other serum or plasma measures reflect the dietary intakes of the past few hours (triglyceride) or the past few days (cholesterol ester and phospholipid fatty acids).'. In addition, he had written 'The next most immediate biomarker medium is the serum or plasma levels of individual fatty acids, which can reflect intake over the last few days or meals.' For your knowledge, check this paper: "Am J Clin Nutr 2007;86:74–81".

Re: Thank you for this comment and the reference. We have revised our manuscript to indicate that plasma phospholipid PUFA levels reflect the dietary intake of the past few days given we have measured the phospholipid fatty acids. 

Lines 361-365, the observations about fasting and non-fasting condition in subjects analyzed is not completely supported by reference [48]. In the cited paper the authors compare lipid in the fasting state and after 1 or 2 hours after breakfast, but in your works the biospecimens preparation, in particular, the time of last meal of subjects is not provided in the sections 2.3. The types of meal (breakfast or lunch or dinner) are very different in the amount and composition of lipids. Also, 2 hours is too less to observe a change in lipid profile after a meal, because the lipid digestive process lasts 4-6 hours approximately.

Re: Thank you for this comment. We have provided the average time interval between the last meal and the blood collection in the revised manuscript (page 3). The mean time interval for visit 0, 1 and 4 were 3.68, 3.19, and 2.90 hours, respectively. We also examined the associations of the time interval with individual PUFAs at all 3 visits and didn’t find any significant associations (original manuscript page 13, line 362-363). We have added a new supplement table 4 in the revised manuscript to show that the study finding remains unchanged with additional adjusting for the time interval. For example, the regression coefficient of LA (Table 2) was -0.287 (P<0.001) before adjusting for the time interval and was -0.280 (P=0.001) after adjusting for time interval. Thus, our analyses suggested the time interval between the last meal and the blood collection was not a real confounder for the association between MVPA and PUFAs. Nevertheless, we have removed the reference # 49 and acknowledged that future studies are needed to confirm our results.

In the discussion highlight that estimation of the plasma delta-5-desaturase (AA/DGLA) and delta-6-desaturase (GLA/LA) activities using the product-to-precursor ratios is a study limit, because not exclude the effects of dietary intake in the previous day before collecting sample plasma, because 24h dietary recall is not performed at collecting blood time.

Re: Thank you for this comment. We believe the effect of dietary intake in the previous day is not a major concern because have measured the phospholipid fatty acids in this study. The literature has indicated that plasma phospholipid PUFA levels reflect the dietary intake of the past few days.

The discussion would be corrected with the previous suggestions.

Technical additional suggestions:

  1. Please improve the sharpness and quality of Figure 1.

Re: We have improved the quality of figure 1 in the revised manuscript. 

  1. Supplement Table 3 are not present in supplementary material attached.

Re: Thanks for this comment. We have included the Supplement Table 3 in the revised manuscript. 

Reviewer 3 Report

This manuscript addresses the association between Plasma PUFA and maternal physical activity in a pregnancy longitudinal study.  This is a nested case control study of 321 women (107 with GDM, 214 without GDM), who had plasma PUFA measured during pregnancy. The authors concluded that MVPA may be a modifiable risk factor to improve plasma PUFA profile.

I have a few comments:

Line 42,43: “emerging evidence had indicated that maternal and circulation PUFA levels play important role om achieving optimal maternal health and fetal development”.

It would be better to spell out the maternal and fetal outcomes. Is this a positive association?

In line 47-51, authors stating that maternal PUFA level is associated with pregnancy complications? High or low? It is better for the author here to be clear on the exact role of PUFA and maternal / fetal outcomes

Line 84-86: There is no rational for sampling of pregnant women (107 GDM and 215 with no GDM). The objective of the study is exploring relationship of maternal PA and maternal individual PFUA. Having 1/3 of the sample women with GDM can over represent pts with metabolic syndrome and overestimate the association.

Author Response

This manuscript addresses the association between Plasma PUFA and maternal physical activity in a pregnancy longitudinal study.  This is a nested case control study of 321 women (107 with GDM, 214 without GDM), who had plasma PUFA measured during pregnancy. The authors concluded that MVPA may be a modifiable risk factor to improve plasma PUFA profile. I have a few comments: 

Line 42,43: “emerging evidence had indicated that maternal and circulation PUFA levels play important role om achieving optimal maternal health and fetal development”. It would be better to spell out the maternal and fetal outcomes. Is this a positive association?

Re: Thanks for this comment. Since the directions of the associations between PUFAs and pregnancy outcomes are varied by individual PUFAs, we have provided some examples in the revised manuscript (page 2, line 51-53).

“For example, a positive association between maternal plasma level of n-6 DGLA and GDM, but an inverse association between n-6 DTA and GDM was observed.  For the fetal outcomes, maternal levels of n-3 DHA and EPA were positively associated with birth weight, whereas n-6 DGLA and AA were inversely associated with birth weight”.

In line 47-51, authors stating that maternal PUFA level is associated with pregnancy complications? High or low? It is better for the author here to be clear on the exact role of PUFA and maternal / fetal outcomes

Re: Thanks for this comment. We have provided different examples for the maternal outcomes and fetal outcomes in the revised manuscript (page 2, line 51-53). It is likely the individual PUFAs have different roles on different maternal and fetal outcomes.

Line 84-86: There is no rational for sampling of pregnant women (107 GDM and 215 with no GDM). The objective of the study is exploring relationship of maternal PA and maternal individual PFUA. Having 1/3 of the sample women with GDM can over represent pts with metabolic syndrome and overestimate the association.

Re: Thank you for this comment. This is a secondary analysis using a unique existing study with longitudinal data on maternal physical activity and plasma PUFAs. To account for the nested case-control design of the 1:2 matching on GDM status, we applied sampling weights to all statistical analyses in current manuscript (original submission page 3, line 131). Thus, the results from our analyses should represent the full NICHD Fetal Growth Studies-Singletons study population. Additionally, we conducted sensitivity analyses among women without GDM and confirmed the findings from the main analyses were unchanged ((original submission page 8, line 234-235 and supplement table 2). We have provided detailed information about the sampling weights method (see below) in the revised manuscript. (page 3)

“The aim of current study was to examine the association between maternal PA and circulating levels of phospholipids PUFAs using biomarker data measured as part of a nested GDM case-control study. Because women with GDM were overrepresented in the analytic sample, we applied sampling weights to all analyses to represent the full NICHD Fetal Growth Studies-Singletons population and account for the oversampling of women with GDM. Weights were created using the inverse probability of each subject in the full cohort (i.e., sampling probability of each non-GDM subject was calculated from a logistic regression in the full cohort, excluding GDM cases). In the weighted sample 4% of women had GDM as opposed to 33% in the non-weighted sample.”

Round 2

Reviewer 3 Report

Thanks for responding to the comments.